# Fibrinolytic Dysregulation in Regional Hemostasis During Liver Transplantation: A Viscoelastometry-Based Pilot Study

**DOI:** 10.3390/jcm14092925

**Published:** 2025-04-24

**Authors:** István Zátroch, Elek Dinya, Anikó Smudla, János Fazakas

**Affiliations:** 1Doctoral College, Semmelweis University, 1085 Budapest, Hungary; zatroch.istvan@gmail.com; 2Institute of Digital Health Sciences, Semmelweis University, 1094 Budapest, Hungary; dinya.elek@public.semmelweis-univ.hu; 3Department of Surgery, Transplantation and Interventional Gastroenterology, Semmelweis University, 1085 Budapest, Hungary; aniko_smudla@yahoo.com; 4Department of Anesthesiology and Intensive Therapy, Semmelweis University, 1085 Budapest, Hungary

**Keywords:** viscoelastometry, liver transplantation, regional hemostasis, systemic hemostasis, blood coagulation, fibrinolysis, ClotPro^®^

## Abstract

**Background/Objectives**: In chronic liver disease, a rebalanced coagulation state often results in an increased risk of thrombosis, particularly in the splanchnic region. While systemic coagulation abnormalities are well documented, alterations in regional (portal) hemostasis remain underexplored. This study aimed to compare systemic and portal hemostasis during liver transplantation and to determine whether systemic parameters can accurately predict regional coagulation status. **Methods**: Thirty-five liver transplant recipients were included in this study. Systemic blood samples (S1–S5) were collected from the external jugular vein at five surgical time points, while portal blood samples (R3) were obtained immediately before reperfusion simultaneously with S3. All samples were analyzed using ClotPro^®^ viscoelastic assays, conventional coagulation tests, and blood gas analysis. **Results**: The EX-test comparison between S3 and R3 samples revealed a discrepancy between systemic and regional hemostasis in 45.7% of patients. Among these, eight regional samples exhibited hypocoagulation characterized by coagulation factor consumption and hyperfibrinolysis. Another eight samples demonstrated hypercoagulation with fibrinolytic shutdown, which was confirmed by a fibrin-rich thrombus identified via scanning electron microscopy. Systemic samples failed to predict these regional variations. **Conclusions**: Regional (portal) hemostasis significantly differs from systemic coagulation and cannot be accurately predicted using systemic assays alone. These findings suggest that fibrinolytic shutdown in the portal vein may contribute to intraoperative and long-term graft damage, highlighting a potential need for regional coagulation assessment during liver transplantation.

## 1. Introduction

Chronic liver disease results in a complex and unstable balance of coagulation processes, often shifting toward thrombosis or hemorrhage. This delicate equilibrium with decreased factor reserves particularly becomes apparent in the splanchnic region, where portal vein thrombosis is a frequent complication as the liver disease progresses. While systemic coagulation abnormalities are well documented, regional hemostasis, especially within the portal vein, remains poorly understood.

Liver transplantation offers a unique setting to compare systemic and regional hemostatic mechanisms. During transplantation, the anhepatic phase and subsequent reperfusion induce profound alterations in coagulation. Current hemostasis monitoring solely relies on systemic blood samples, which may not accurately reflect the regional conditions within the portal vein.

Recent studies have highlighted the potential role of viscoelastometry in assessing coagulation abnormalities; however, comparative investigations of regional (portal) and systemic coagulation during liver transplantation are lacking.

The present study aims to address this gap by comparing systemic and portal hemostasis during liver transplantation using ClotPro^®^ viscoelastometry. The primary goal is to determine whether systemic parameters can reliably predict portal hemostatic conditions and to identify specific characteristics of alterations in blood coagulation that may potentially contribute to intraoperative graft damage.

## 2. Patients and Methods

This prospective, single-center, observational study was conducted at the Department of Transplantation and Surgery, Semmelweis University, Budapest, Hungary, which is the sole center in Hungary conducting liver transplantation. The study has received a positive opinion from the Hungarian Ethics Committee Medical Research Council (20325-2/2019/EKU) and is registered on ClinicalTrials.gov (NCT0424637). The research was conducted under the principles of the Helsinki Declaration. Written informed consent was obtained from each participant before enrollment.

Thirty-five adult patients with chronic liver disease undergoing liver transplantation from a cadaveric donor participated in the study. We used the following exclusion criteria for the study: patients under 18 years of age, those with acute liver failure, individuals requiring multiple organ transplants, patients needing retransplantation, and those who did not provide written informed consent.

Systemic blood samples were collected at five different time points (S1, S2, S3, S4, and S5) during surgery from a dedicated cannula inserted into the external jugular vein specifically for coagulation sample collection. The sampling times were as follows: before surgery (S1), during hepatectomy 10 min before the anhepatic phase (S2), during the anhepatic phase when the portal vein was being anastomosed (S3), during the neohepatic phase 15 min after reperfusion of the portal vein (S4), and at the end of the surgery (S5). In addition, regional samples (R3) were obtained from the portal vein simultaneously with the S3 sample (Figure 1). Two 3.5 mL Vacuette blood collection tubes (Greiner Bio-One International GmbH, Kremsmünster, Austria) containing 3.2% sodium citrate and one 3.5 mL potassium ethylenediamine tetraacetic acid (K2EDTA) tube were used for each sample. Blood gas analysis was conducted from both systemic and regional samples to investigate the hemostatic environment (Figure 2).

Viscoelastometric assays were performed using a ClotPro^®^ analyzer (Haemonetics, Boston, MA, USA). The ClotPro^®^ features an innovative measurement technique that enables a rapid and dynamic evaluation of blood coagulation using whole blood. As in the original Hartert method, the pin remains stationary in this equipment and the cup is rotated by a flexible element. To reduce technical errors, the device employs an “active pipette tip”, which contains dried reagents embedded in a sponge at the pipette’s tip. For each assay, 340 µL of sodium citrate-treated blood is pipetted into the cuvette using the active tips. The coagulation process is initiated by the various activators. The elastic element rotates the cup, stretching in proportion to the size of the clot formed. The device displays this stretching both graphically and quantitatively over time.

The ClotPro^®^ can perform six assays simultaneously. In this study, EX-, FIB-, IN-, TPA-, RVV- and ECA-tests were applied simultaneously. The EX-test allows the assessment of the extrinsic pathway of blood coagulation. It is activated by a recombinant tissue factor that is recalcified with calcium chloride, while the FIB-test, due to containing thrombocyte inhibitors (cytochalasin-D and abciximab), measures the amount of functional fibrinogen after tissue factor activation. The TPA-test is also initiated by tissue factors on the extrinsic pathway and includes a recombinant tissue plasminogen activator to evaluate antifibrinolytic activity. After recalcification, ellagic acid in the IN-test initiates the intrinsic pathway of coagulation. The RVV-test contains the Russell’s viper’s (Daboia russelii) venom, a direct factor X activator, which is used to monitor inhibitors of activated factor X. The ECA-test contains ecarin, which is a substance extracted from the venom of the snake Echis carinatus that was developed to monitor the effect of thrombin antagonists (Appendix A). Each testing session was 60 min in duration.

The clotting time (CT), time to clot formation (CFT), maximal clot firmness (MCF), maximal clot lysis (ML) and, for the TPA-test, clot lysis time (LT), were recorded for each assay. CT is defined as the time elapsed from the start of the testing process until the clot reaches a size of 2 mm. CFT means the time elapsed until the amplitude reaches 20 mm. MCF is determined as the maximum amplitude measured during the assay. ML is defined as the maximum decrease in the amplitude over one hour after reaching the MCF, while LT is the time elapsed until the amplitude decreases to 50% of the MCF after reaching CT (Appendix A).

In addition to viscoelastic assays, the conventional coagulation parameters (INR, aPTT, and D-dimer), coagulation factor activity (FV, FVII, FX, and FXIII) and platelet count (PLT) were determined using Sysmex CS 2000i and Sysmex XN-100 instruments (Sysmex Europe SE, Norderstedt, Germany). (Figure 3). Blood gas analysis was conducted using a GEM Primer 3500 equipment (Werfen, Bedford, MA, USA). In addition, a scanning electron microscopy (SEM) analysis of a thrombus formed in a sodium citrate tube from the portal vein was performed using a Zeiss EVO 40 appliance (Carl Zeiss Meditech AG, Jena, Germany). Furthermore, for the detection of fibrin, extracellular DNA and neutrophils, an immunofluorescence assay was conducted.

Statistical analysis: Continuous variables were analyzed using descriptive statistics. The Shapiro–Wilk test was employed to assess normality, while Levene’s test was used to evaluate the homogeneity of variances. Since the assumptions for parametric ANOVA were not met, the differences between groups were analyzed using the non-parametric Kruskal–Wallis (ANOVA) test. Additionally, Wilcoxon’s test was applied to compare systemic (S3) and regional (R3) samples. A result was considered statistically significant if *p* < 0.05. Statistical analyses were performed using IBM SPSS Statistics 28.0 for Windows (SPSS Inc. Chicago, IL, USA) software.

## 3. Results

The study involved 35 patients with an average age of 48.7 ± 13 years and with a predominance of male sex (65.7%). The most common indications for liver transplantation were cirrhosis of biliary origin (34.3%) and alcoholic liver disease (25.7%). Two thirds of patients were classified in Child–Turcotte–Pugh groups B and C with a median MELD-Na score of 14 points (Table 1).

Using the EX-test, which allows a comprehensive evaluation of the extrinsic coagulation pathway, an analysis of systemic (S3) and regional (R3) blood samples collected before portal reperfusion indicated a difference in coagulation status between systemic and regional samples in nearly half of the patients (45.7%). The results of the EX-test assay revealed three distinct groups of patients. In 19 patients, the coagulation in both systemic and regional samples was identical. Meanwhile, eight patients exhibited hypocoagulation, and another eight individuals showed hypercoagulation in the portal samples (R3) compared to the systemic samples (S3) as illustrated in Figure 4. The three groups had no significant differences concerning demographic data or the intraoperative surgical course. Of the 35 transplantations, 77% (27/35) were performed using the classic cross-clamp technique, and 23% (8/35) were performed using the piggyback technique. No portocaval shunt was used. Intraoperative blood product administration was comparable among the three fibrinolytic groups. There were no significant differences in the mean doses of fibrinogen concentrate, prothrombin complex concentrate, red blood cells, fresh frozen plasma or platelet transfusion.

In the hyperfibrinolysis group, two patients required reoperation for the evacuation of perihepatic hematomas. In the group with identical fibrinolytic profiles, three hematoma evacuations and two hepatic artery thrombectomies were performed. In contrast, the fibrinolysis shutdown group did not undergo any bleeding-related reinterventions; however, two portal vein thrombectomies and one hepatic artery thrombectomy were necessary in this group (Table 1).

There was no statistically significant difference among the three groups regarding the investigation of systemic samples (S1, S2, S4, and S5) using viscoelastometric assays, conventional laboratory tests, and blood gas analysis. The pre-reperfusion systemic samples (S1 and S2) did not predict the coagulation characteristics of the portal sample, and similarly, subsequent systemic samples (S4 and S5) failed to distinguish between the three groups (Appendix A).

Portal samples showing hypocoagulation via the ClotPro^®^ EX-test demonstrated significantly lower values of prothrombin time (*p* = 0.028), fibrinogen level (*p* = 0.027), FV (*p* = 0.028) and FXIII (*p* = 0.018) activities along with an elevated D-dimer level (*p* = 0.018). Although the decline in calcium (*p* = 0.1) and platelet (*p* = 0.12) counts did not meet the assumption of significance, they showed remarkable decreases (Figure 5, Appendix A). Overall, these findings suggest that this subgroup is characterized by hyperfibrinolysis.

One of the hypercoagulable samples that formed a clot despite the presence of sodium citrate in the cuvette was subjected to further analysis by scanning electron microscopy. The images revealed patchy fibrin fibers of varying thickness, a limited number of platelets, and biconcave-shaped red blood cells. These observations may suggest an increased tendency toward coagulation and the formation of a resistant thrombus (Figure 6). The immunofluorescence assay indicated a fibrin-rich thrombus, yet extracellular DNA and neutrophils could not be identified. In brief, these findings suggest that this subgroup is characterized by fibrinolysis shutdown.

## 4. Discussion

In chronic liver disease, impaired synthetic and degradative functions of the liver result in an altered, unstable balance in blood coagulation processes with reduced hemostatic reserves [1]. Any shift in this delicate balance, either toward thrombosis or hemorrhage, implies severe clinical consequences. As the liver failure progresses, there is an increasing tendency for thrombosis in the splanchnic region. In cirrhosis, portal venous thrombosis occurs in 8–25% of cases, the prevalence increasing with disease progression. In contrast, this rate in the general population is only ca. 1% [2]. Portal vein thrombosis after liver transplantation is a rare but serious complication [3]. Occasional microthrombus formation at the conclusion of the anhepatic phase and its subsequent drift into the graft with the portal reperfusion can pose a major challenge. Owing to reduced fibrinolytic activity or fibrinolysis shutdown, these thrombi may lead to centrolobular ischemia, which can indirectly damage the biliary epithelium. This injury may result in long-term biliary tract complications or even progressive graft damage. Therefore, in clinical decision making, it may be relevant to test regional blood samples.

Vicoelastometry has been applied since the early days of liver transplantation surgery because it offers whole blood-based rapid and comprehensive assessment of the dynamic changes in coagulation processes that occur during the procedure [4,5,6,7,8]. The delivered information contributes to reducing the need for allogeneic blood product transfusions without increasing the rate of complications [9,10,11,12]. The clot size in the assay strongly correlates with a hypercoagulable state [13,14,15,16]. The various viscoelastic test reagents allow for evaluating different aspects of blood coagulation. Together, they provide a detailed and comprehensive overview of hemostasis. For instance, the TPA-test investigates antifibrinolytic processes, while the ECA-test can be used to detect increased fibrinolytic activity earlier [17,18]. Considering all these factors, we decided to proceed with ClotPro^®^ for blood sample analysis. To our knowledge, no study has yet compared systemic and portal coagulation using viscoelastometry alongside the evaluation of the underlying plasmatic processes through conventional coagulation laboratory parameters and the assessment of the hemostatic environment via blood gas analysis.

Although two surgical techniques were applied (classic and piggyback), their distribution across fibrinolytic subgroups was relatively balanced, and no clear correlation was observed between surgical approach and the fibrinolytic phenotype.

The presented preliminary results suggest that systemic and regional coagulation parameters before portal reperfusion may show substantial variations during liver transplantation. In nearly half of the patients, portal and systemic coagulation parameters were inconsistent, which resulted in the identification of three distinct groups: those with identical coagulation values in both the systemic and portal samples, those showing a tendency to hyperfibrinolysis, and those exhibiting a tendency toward fibrinolysis shutdown in the portal samples.

This finding aligns with earlier studies highlighting individual features of regional hemostasis in chronic liver disease. Rossetto et al. [19] emphasize the unique coagulation profiles observed in the portal venous system. Indeed, our results supported the notion that monitoring portal and systemic coagulation separately during surgery may be beneficial to facilitate successful transplantation and prevent complications.

Thanks to advancements in surgical and anesthetic techniques, liver transplantation can now be performed without blood product administration, including red blood cells, plasma, or other coagulation products [20,21]. In our cohort, approximately half of the patients in each fibrinolytic group underwent factor concentrate and blood-product-free transplantation. No cases of massive transfusion occurred in our cohort.

The ClotPro^®^ EX-test assay revealed significantly lower fibrinogen, FV, and FXIII levels in the hypocoagulated portal samples, which were associated with elevated D-dimer values. A decrease in the calcium levels and platelet counts was observed. Although this decrease did not reach statistical significance, the trend was consistent with the characteristics of hypocoagulation. Eventually, these findings characterize this group as having a concomitant consumption of clotting factors and overactivated fibrinolysis.

Examination of a blood clot in a sodium citrate-resistant hypercoagulable sample using scanning electron microscopy (SEM) revealed a hypocellular fibrin-rich thrombus, which is consistent with the observations of Driever et al. in their study [22]. Thrombi that form in the venous system are typically characterized by low platelet content, as noted in the analysis of the SEM image as well. This low platelet count likely explains why the red blood cells maintained their biconcave shape and did not transform into polyhedrocytes [23]. On the basis of the results of the immunofluorescence assay, which did not detect any extracellular DNA or neutrophils, the possibility of immunothrombosis can be ruled out.

The most important factor influencing fibrin fiber size is thrombin activity. At high thrombin concentrations, a dense network of thin, highly branched fibrin fibers is formed, making the clot more resistant to fibrinolysis [24,25,26]. In the clot analyzed in our study, we observed fibrin fibers of varying thickness and density, suggesting patchy and heterogeneous thrombin generation. Some areas showed dense fibrin networks with numerous connections. These findings may suggest that this group is also characterized by the consumption of clotting factors; however, in contrast to the hypocoagulable group, low fibrinolytic activity could be detected in the sample.

Additionally, a reduction in calcium levels observed in the blood gas analysis may hold further implications. Since regional blood gas analysis can be performed immediately during surgery, it would be of interest to explore its potential to differentiate between fibrinolytic phenotypes. In our cohort, calcium levels remained unchanged in the group with identical portal and systemic parameters, showed a marked decrease in the hyperfibrinolysis group, and spontaneous clotting of the sample occurred in the fibrinolysis shutdown group.

According to Virchow’s triad, clot formation is promoted by stasis, a hypercoagulable state, and endothelial injury. In this study, reduced blood flow was a common feature of all groups during the anhepatic phase, and systemic parameters showed no differences. Therefore, the differing local behavior of endothelial cells might be responsible for the observed variations. It is important to emphasize that the interpretations above are hypothetical and based on morphological and laboratory findings rather than clinical outcomes.

The findings of Driever et al. [22] support our hypothesis. In their study, 79 portal vein thrombi collected during liver transplantation are analyzed using scanning electron microscopy and immunohistochemistry. All samples show a thickened, fibrotic tunica intima, and in one third of the cases, a fibrin-rich thrombus was observed. The authors of the present study propose endotheliopathy as the underlying cause of the observed regional results, which actually identify two manifestations of disseminated intravascular coagulopathy: the fibrinolytic and the thrombotic forms [27].

The presented preliminary results suggest that monitoring both portal and systemic hemostasis separately during liver transplantation could help prevent complications because regional coagulation processes cannot be predicted through systemic blood samples alone. However, it is worth noting that abnormalities identified by the ClotPro^®^ EX-test are not specific to any single test. The observed decrease in fibrinogen, factor V, and factor XIII levels in the hypocoagulation group suggests that some patients may face a higher risk of bleeding. If these issues are detected early, targeted treatment could potentially reduce bleeding risk. Nearly one quarter of the participants exhibited fibrinolysis shutdown in the portal samples, which may potentially lead to intraoperative and long-term graft damage. In these cases, managing the risk of thrombosis may become a top priority. This could involve, for instance, administering tissue plasminogen activator (tPA) locally or introducing anticoagulant therapy such as sodium–heparin and antithrombin III. Although antithrombin III (AT III) measurement is not routinely performed in all centers, its assessment may be valuable in selected high-risk cases. In our institution, AT III supplementation is routinely performed when indicated with a target activity level above 80%. This is considered important due to the broad inhibitory effect of AT III on the intrinsic coagulation pathway, including factors XII, XI, IX, X, and II, as well as its essential role as a cofactor for heparin. A recent study evaluating factor concentrates in blood-product-free liver transplantation found that even with supplementation, approximately 24–48 h is needed to restore preoperative AT III levels postoperatively [20]. Furthermore, a retrospective study by Kuznetsova et al. demonstrated that the perioperative administration of AT III concentrate led to a faster normalization of AT III activity, reduced rates of thrombotic and infectious complications, and improved survival [28]. These findings underscore the potential value of individualized anticoagulant strategies based on intraoperative regional hemostatic assessment.

No thrombotic complications were observed in the hyperfibrinolysis group, only hematomas. In the normal fibrinolysis group, both bleeding events and hepatic artery thromboses occurred. In contrast, the fibrinolysis shutdown group exhibited no bleeding events, but two portal vein thromboses and one hepatic artery thrombosis were recorded in this group. While no statistical comparison was possible due to the low number of events, these findings may support the hypothesis that regional fibrinolytic phenotypes influence early vascular complications and warrant further investigation in larger cohorts.

The clinical significance of this study is that viscoelastometric assays enable an accurate, rapid, and dynamic assessment of blood coagulation. This allows for interventions tailored to the patient’s needs. By using ClotPro^®^’s viscoelastometric assays, it is possible to detect an increased tendency to bleed or a risk of thrombosis early on. The continuous monitoring of coagulation during liver transplantation can facilitate timely interventions, ultimately improving patient outcomes.

The authors declare several limitations to the study. Firstly, the relatively small sample size of 35 patients raises concerns about the generalizability of the findings. Apart from this, the study was conducted at a single center, which is a condition that may be an additional limitation for the global applicability of the results. Ultimately, the technical challenges related to portal blood sampling may have impacted the results. Accurately measuring portal blood clotting under surgical conditions is challenging, and these technical difficulties may have influenced the findings.

Despite the disclosed limitations, our findings provide valuable insights into the regional coagulation dynamics in liver transplantation, warranting further investigation in larger cohorts. In addition, the impact of fibrinolytic dysregulation within the portal vein on intraoperative and postoperative graft outcomes remains unclear. Further research may also enhance our understanding of the specific aspects of regional coagulation and their impact on liver transplantation outcomes.

## Figures and Tables

**Figure 1 jcm-14-02925-f001:**
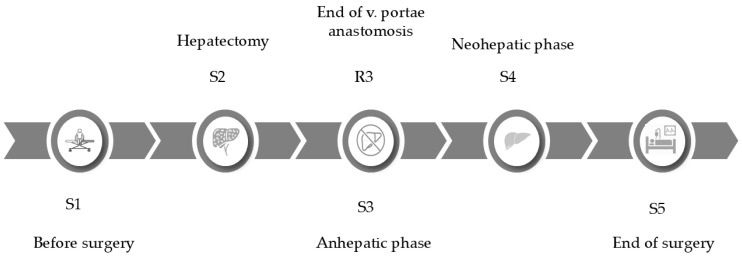
Sampling times. Systemic samples were collected from the external jugular vein: before surgery (S1), during hepatectomy, 10 min before the anhepatic phase (S2), in the anhepatic phase during the preparation of the portal vein anastomosis (S3), in the neohepatic phase, 15 min after reperfusion of the portal vein (S4), and at the end of the surgery (S5). A regional sample (R3) was taken from the portal vein simultaneously with the S3 sample.

**Figure 2 jcm-14-02925-f002:**
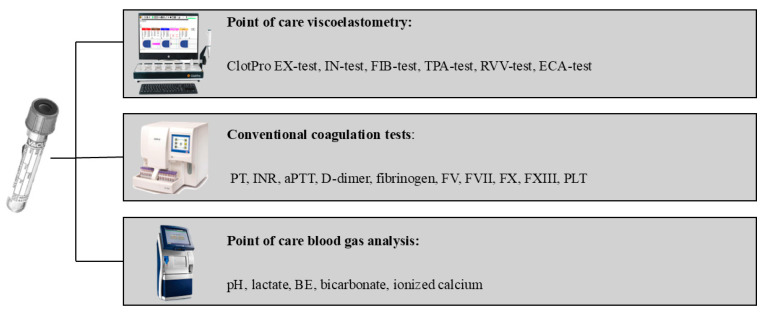
Laboratory investigations from systemic and regional blood samples. aPTT: activated partial thromboplastin time, BE: base excess, FV: factor V, FVII: factor VII, FX: factor X, FXII: factor XIII, INR: international normalized ratio, PLT: platelet, PT: prothrombin time.

**Figure 3 jcm-14-02925-f003:**
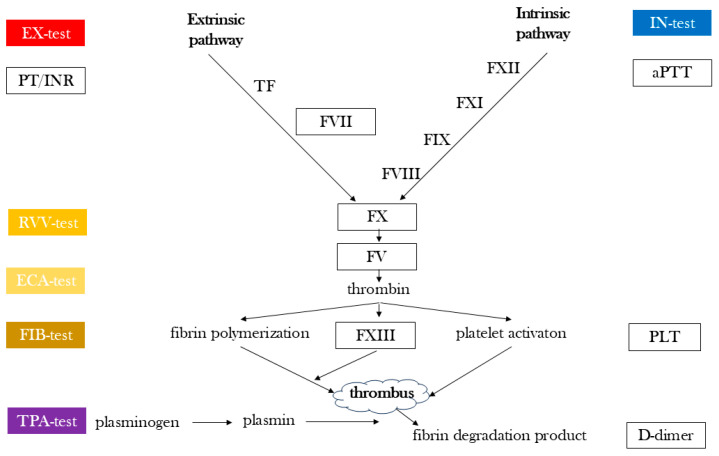
Simplified schematic of the coagulation cascade highlighting the components assessed in the present study. The figure shows the intrinsic, extrinsic, and common pathways with boxed areas indicating the laboratory and viscoelastic parameters (ClotPro^®^ tests and conventional assays) used to evaluate specific parts of the coagulation system. aPTT: activated partial thromboplastin time, PT/INR: prothrombin time/international normalized ratio, F: factor, PLT: platelet.

**Figure 4 jcm-14-02925-f004:**
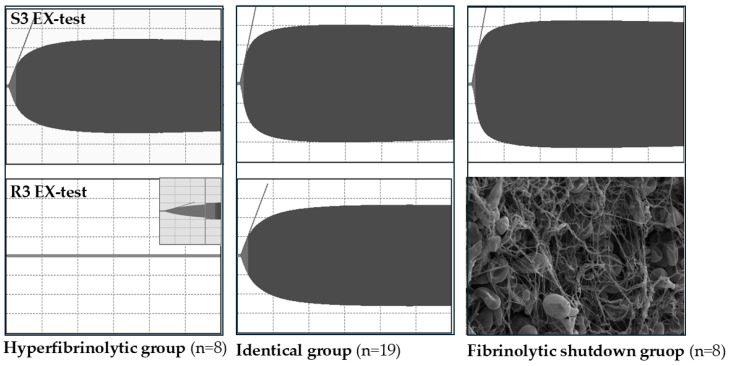
Comparison of systemic (S3) and regional (R3) samples collected simultaneously at the end of the anhepatic phase using the ClotPro EX test.

**Figure 5 jcm-14-02925-f005:**
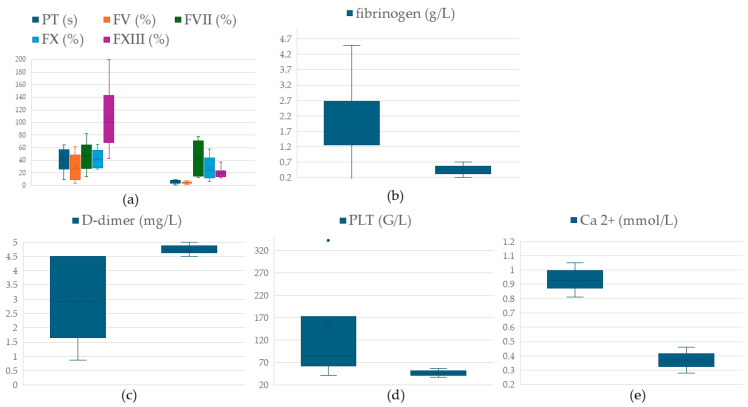
Comparison of identical (left side of each graphic) and hyperfibrinolytic (right side of each graphic) group R3 samples. Significantly lower PT, FV, FXIII, fibrinogen (**a**,**b**), and elevated D-dimer (**c**) are observed along with a notable but not significant decrease in platelet count (**d**) and calcium level (**e**). PT: prothrombin time, FV: factor V, FVII: factor VII, FX: factor X, FXIII: factor XIII, PLT: platelet, Ca^2+^: ionized calcium.

**Figure 6 jcm-14-02925-f006:**
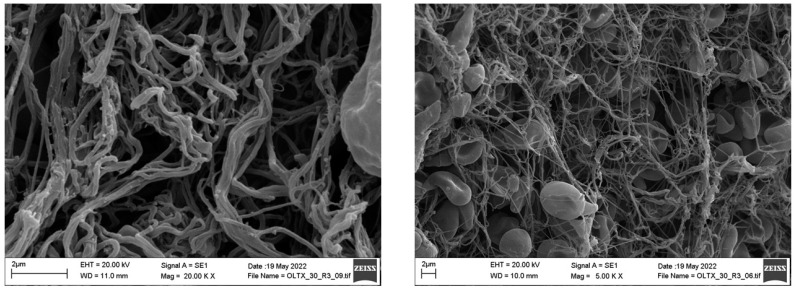
Scanning electron microscopic images of a portal vein sample.

**Table 1 jcm-14-02925-t001:** Demographic and surgical data for the entire study population and the subgroups.

	Patient Number (n = 35)	Hyperfibrinolysis in Portal Vein Sample (n = 8)	Identical Coagulation in Portal and Systemic Blood (n = 19)	Fibrinolysis Shutdown in Portal Vein Sample (n = 8)	*p*
Age (y)	48.7 ± 13	46.9 ± 12.3	50.4 ± 14.3	46.4 ± 11.4	0.565
Sex—man	23/35	2/8	14/19	7/8	
**Aetiology of ESLD**					
Viral hepatitis	5	0	5	0	
Biliary cirrhosis	12	3	4	4	
Alcoholic liver disease	9	2	4	3	
Miscellaneous	9	2	6	1	
**Severity of ESLD**					
CTP—A	9	2	5	2	
CTP—B	22	4	13	5	
CTP—C	4	2	1	1	
MELD-Na score (median)	14	14	14	14	0.861
**Data of surgery**					
Surgery technique (cross-clamping/piggyback)	27/8	6/2	13/6	8/0	
Duration of operation (min)	235.4 ± 56.7	230.2 ± 58.7	231.5 ± 59.4	249 ± 53	0.787
Hepatectomy (min)	69.4 ± 21.3	66.5 ± 17.9	70.8 ± 21.1	68.9 ± 26.8	0.977
Anhepatic phase (min)	57.4 ± 20.4	61.5 ± 24.3	55.5 ± 22	57.7 ± 12.6	0.607
Cold ischemic time (min)	487.2 ± 117.8	485.7 ± 88.9	505.5 ± 130.8	444 ± 112.2	0.368
Warm ischemic time (min)	52.8 ± 20.7	61.1 ± 29.3	50.4 ± 19.7	50.5 ± 11.3	0.76
Reoperation due to bleeding		2	3	0	
Reoperation due to thrombosis		0	2 (HAT)	3 (2 PVT, 1 HAT)	
**Intraoperative factor concentrate and blood component therapy**					
FC (g)		2.7 ± 4.7	2.45 ± 3.5	1.25 ± 1.8	0.613
PCC (U)		285.7 ± 487.9	0	125 ± 353.3	0.066
RBC (U)		2.6 ± 4.4	2.1 ± 2.6	2.1 ± 2.7	0.901
FFP (U)		1.4 ± 3.8	1.6 ± 3.0	0.75 ± 2.12	0.462
PLT conc. (U)		3.1 ± 4.3	1.6 ± 3.5	0	0.134
Factor concentrate and blood product-free (n)		4	8	3	

CTP: Child–Turcotte–Pugh score, MELD: model for end-stage liver disease, RBC: red blood cells, FFP: fresh frozen plasma, FC: fibrinogen concentrate, PCC: prothrombin complex concentrate, U: unit, HAT: hepatic artery thrombosis; PVT: portal vein thrombosis.

## Data Availability

The data presented in this study are available from the corresponding author upon request.

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
