# Peer review of "Fibrinolytic Dysregulation in Regional Hemostasis During Liver Transplantation: A Viscoelastometry-Based Pilot Study"

_jcm, 2025, doi:10.3390/jcm14092925_

Round 1
Reviewer 1 Report
Comments and Suggestions for Authors
- At the outset, there is less overlap in the text confirmed by iThenticate of 18% which is very commendable!
– the research is interesting and encouraging from the perspective of a clinician dealing with transplantation and graft patient outcomes!
The SEM images are truly impressive!
Figure 1. Substituting APTT for aPTT!
- Can antithrombin III determination be considered routine?
-Some practical answers that are useful to transplant clinicians are missing:
- Is it necessary to administer higher doses and fractionated heparins preoperatively?
- How to adjust the heparin dose in simultaneous liver-kidney transplantation?
- Since the time of cold ischemia is not important, can someone explain in more detail?
- Are DSA determinations important?
- Is the underlying liver disease important?
- Are there differences in infectious and non-infectious forms of the disease?
- Are there data in Tx kidney or ESKD?
The entire paper is very correctly written and I have no major complaints
Perhaps a schematic representation of the coagulation cascade is missing in this specific problem!
Author Response
Thank you very much for taking the time to review this manuscript. Please find the detailed responses below and the corresponding revisions in track changes in the re-submitted files.
Comments 1: At the outset, there is less overlap in the text confirmed by iThenticate of 18% which is very commendable! The research is interesting and encouraging from the perspective of a clinician dealing with transplantation and graft patient outcomes! The SEM images are truly impressive!
Response 1: Thank you for your positive and encouraging comments regarding the originality of the manuscript, the quality of the SEM images, and the clinical relevance of the research. We are grateful for your supportive evaluation.
Comments 2: Figure 1. Substituting APTT for aPTT!
Response 2: Thank you for pointing this out. We have corrected the notation in Figure 1 and throughout the manuscript to use the appropriate abbreviation "aPTT".
Comments 3: Can antithrombin III determination be considered routine?
Response 3: We thank the reviewer for this important question. AT III deficiency—common in end-stage liver disease—can impair heparin efficacy. While antithrombin III (AT III) determination is not considered routine in all liver transplantation centers, its clinical relevance is increasingly recognized, especially in high-risk patients or in cases where heparin resistance due to low levels of AT III is suspected. As highlighted by Kuznetsova et al. (2019), perioperative AT III monitoring and supplementation may reduce thrombotic complications and improve outcomes. We have included a short discussion of this point in the revised manuscript (see Discussion, lines 340-353).
Comments 4: Is it necessary to administer higher doses and fractionated heparins preoperatively?
Response 4: We thank the reviewer for this clinically relevant question. Administering higher doses of heparin preoperatively could carry an unnecessary bleeding risk. There is currently no standardized protocol for preoperative administration in liver transplantation. In our current study, patients did not receive preoperative anticoagulation with heparin. Although higher or fractionated doses of heparin might be considered in high-risk cases. Importantly, based on systemic coagulation parameters alone, it is not possible to identify patients with portal hypercoagulability prior to surgery. However, if intraoperative sampling reveals signs of portal hypercoagulability, targeted anticoagulation with sodium heparin—potentially at higher doses—may be justified in this subgroup.
Comments 5: How to adjust the heparin dose in simultaneous liver-kidney transplantation?
Response 5: Although our study did not include patients undergoing simultaneous liver-kidney transplantation (SLKT), the topic of anticoagulation management in this setting deserves specific attention. SLKT patients often present with complex and unstable coagulation profiles due to the combined effects of liver dysfunction, renal failure, and uremia-related platelet dysfunction. In such cases, both thrombotic and bleeding risks are elevated, and heparin dosing must be carefully adjusted. In our institution, patients undergoing simultaneous liver-kidney transplantation (SLKT) receive heparin and antithrombin III supplementation after portal reperfusion, during the kidney implantation phase. This approach is intended to reduce the risk of thrombotic complications, especially in patients with impaired endogenous anticoagulant capacity. The heparin dose is adjusted based on laboratory results, and AT III is administered with the goal of maintaining activity levels above 80%.
Comments 6: Since the time of cold ischemia is not important, can someone explain in more detail?
Response 6: We thank the reviewer for raising this important point. While CIT may play a role in post-reperfusion coagulation disturbances or endothelial activation. In our study, regional blood samples was collected during the anhepatic phase, prior to graft reperfusion. At this time, the donor liver was not yet reconnected to the circulation, and therefore the cold ischemia time (CIT) could not have influenced the measured coagulation parameters.
Comments 7: Are DSA determinations important?
Response 7: We thank the reviewer for this important immunological consideration. Donor-specific antibody (DSA) testing is indeed a key component in the immunological risk assessment of solid organ transplantation, particularly in kidney and combined liver-kidney transplantation. However, our current study focused specifically on hemostatic and fibrinolytic parameters, and we did not include DSA screening or correlation with coagulation profiles. While emerging evidence suggests potential links between immune activation and coagulation imbalance, this relationship remains complex and beyond the scope of our present analysis.
Comments 8-9: Is the underlying liver disease important? Are there differences in infectious and non-infectious forms of the disease?
Response 8-9: We thank the reviewer for this important question. We examined whether the etiology of liver disease influenced the coagulation profiles. No significant differences were observed between these subgroups in terms of viscoelastic parameters or fibrinolytic patterns. The sample size was limited, and the lack of statistical difference may be due to insufficient power. The potential role of liver disease etiology—including infectious vs. non-infectious causes—on coagulation is common sense.
Comments 10: Are there data in Tx kidney or ESKD?
Response 10: We thank the reviewer for this thoughtful and relevant question. End-stage kidney disease (ESKD) is known to affect coagulation through platelet dysfunction, altered fibrinolysis, and reduced clearance of anticoagulants. To our knowledge, no studies have specifically investigated regional coagulation profiles in the renal circulation of kidney transplant recipients.
Comments 11: Perhaps a schematic representation of the coagulation cascade is missing in this specific problem
Response 11: We thank the reviewer for highlighting the need for a visual representation of the coagulation cascade in this context. In response, we created a simplified schematic diagram that illustrates the main coagulation pathways and indicates the specific laboratory and viscoelastic parameters assessed in our study. This figure has been incorporated into the revised manuscript as Figure 3. We are grateful for the reviewer’s constructive comments, which have helped us improve the clarity, depth, and clinical relevance of our manuscript. We hope that the revised version adequately addresses the raised concerns.

Reviewer 2 Report
Comments and Suggestions for Authors
This study explores the differences between systemic and portal coagulation hemostasia at a specific point in the anhepatic phase. An initial R2 extraction should also have been done and compared with S2 before the anhepatic phase to see if the different profile of hyperfibrinolysis and shut down was shown before the anhepatic phase. Limitations in the number of measurements mean that the differences between phases of the systemic coagulation are not significant, they lose value the main objective of the study to observe the repercussions of the regional coagulation (portal) on the systemic and the potential clinical value.
Data that should be provided to make an accurate assessment:
- Surgical technique: preservation of the vena cava, portocaval shunt, both can influence the determinations of systemic coagulation.
- Transfusion of blood component during the procedure
- Bleeding problems and portal or arterial thrombosis in theintraoperative or postoperative period. These may be a manifestation of regional fibrinolysis o shuntdown pathology during the transplant In fact, the authors describe the clinical advantages in relation to fibrinolysis in both directions (lines 275 to 291) without any clinical evidence to support it.
Author Response
Thank you very much for taking the time to review this manuscript. Please find the detailed responses below and the corresponding revisions in track changes in the re-submitted files.
Comments 1: Comments and Suggestions for Authors
This study explores the differences between systemic and portal coagulation hemostasia at a specific point in the anhepatic phase. An initial R2 extraction should also have been done and compared with S2 before the anhepatic phase to see if the different profile of hyperfibrinolysis and shut down was shown before the anhepatic phase. Limitations in the number of measurements mean that the differences between phases of the systemic coagulation are not significant, they lose value the main objective of the study to observe the repercussions of the regional coagulation (portal) on the systemic and the potential clinical value.
Response 1: We thank the reviewer for this thoughtful and constructive suggestion.
We agree that including an additional portal sample prior to liver explantation (R2) would allow for more detailed analysis of fibrinolytic activity and regional hemostasis before the anhepatic phase. While the current study was limited to a single portal sampling point during the anhepatic phase, we fully acknowledge the value of a pre-anhepatic comparison.
We are planning to continue the study with an expanded cohort and will revise the protocol to incorporate the R2 sampling point as recommended.
Comments 2: Surgical technique: preservation of the vena cava, portocaval shunt, both can influence the determinations of systemic coagulation.
Response 2: We thank the reviewer for this important and relevant question.
Among the 35 liver transplantations included in the study, 27 were performed using the classic technique with cross-clamping, and 8 with the piggyback technique; no temporary portocaval shunt was used in any case.
The distribution of surgical technique across fibrinolytic subgroups was as follows:
- hyperfibrinolysis group: 6 classic / 2 piggyback
- normal fibrinolysis group: 13 classic / 6 piggyback
- fibrinolysis shutdown group: 8 classic / 0 piggyback
Although the surgical technique was not entirely uniform, no clear association was observed between the type of technique and the fibrinolytic profile.
We have added this information to the revised manuscript.
Comments 3: Transfusion of blood component during the procedure.
Response 3: We thank the reviewer for highlighting the importance of evaluating transfusion-related variables. We compared intraoperative blood product use across the three fibrinolytic subgroups. We have expanded Table 1 to include recommended information, there were no significant differences in the administration of , fibrinogen, prothrombin complex concentrate, red blood cells, fresh frozen plasma or platelet concentrates.
Comments 4: Bleeding problems and portal or arterial thrombosis in the intraoperative or postoperative period. These may be a manifestation of regional fibrinolysis or shuntdown pathology during the transplant In fact, the authors describe the clinical advantages in relation to fibrinolysis in both directions (lines 275 to 291) without any clinical evidence to support it.
Response 4: We thank the reviewer for this critical and clinically relevant comment.
We reviewed all postoperative imaging (ultrasound, CT) and surgical reports to assess the occurrence of bleeding and thrombotic complications.
In total, 11 reoperations were performed due to suspected hemorrhage or vascular thrombosis.
- In the hyperfibrinolysis group, two reoperations were performed for perihepatic hematoma evacuation.
- In the identical fibrinolysis group, three hematoma evacuations and two hepatic artery thrombectomies were performed.
- In the fibrinolysis shutdown group, no reoperations were performed due to bleeding; however, two portal vein thrombectomies and one hepatic artery thrombectomy were required.
These events were too few for statistical analysis, but may suggest that regional fibrinolytic profiles could be associated with specific vascular complications.
We have added this information and clarification to the revised manuscript.
We have revised the relevant section of the manuscript accordingly and clarified the hypothetical nature of the interpretation.
We are grateful for the reviewer’s constructive comments, which have helped us improve the clarity, depth, and clinical relevance of our manuscript. We hope that the revised version adequately addresses the raised concerns.
